# Provable Efficiency of Guidance in Diffusion Models for General Data Distribution

**Gen Li** [* 1]  **Yuchen Jiao** [* 1]

## Abstract

Diffusion models have emerged as a powerful framework for generative modeling, with guidance techniques playing a crucial role in enhancing sample quality. Despite their empirical success, a comprehensive theoretical understanding of the guidance effect remains limited. Existing studies only focus on case studies, where the distribution conditioned on each class is either isotropic Gaussian or supported on a one-dimensional interval with some extra conditions. How to analyze the guidance effect beyond these case studies remains an open question. Towards closing this gap, we make an attempt to analyze diffusion guidance under general data distributions. Rather than demonstrating uniform sample quality improvement, which does not hold in some distributions, we prove that guidance can improve the whole sample quality, in the sense that the ratio of bad samples (measured by the classifier probability) decreases in the presence of guidance. This aligns with the motivation of introducing guidance.

## 1. Introduction

Score-based diffusion models have recently emerged as an expressive and flexible class of generative models, demonstrating competitive performance on image and audio synthesis tasks (Sohl-Dickstein et al., 2015; Song & Ermon, 2019; Ho et al., 2020; Song et al., 2021b;a; Croitoru et al., 2023; Ramesh et al., 2022; Rombach et al., 2022; Saharia et al., 2022). These models operate through a forward process, which progressively transforms data from the target distribution into Gaussian noise, and a reverse process that

generates samples. The reverse process typically involves approximating the score function—defined as the gradient of the log-likelihood of noisy distributions—at various scales by training a neural network (Hyvärinen, 2005; Ho et al., 2020; Hyvärinen, 2007; Vincent, 2011; Song & Ermon, 2019; Pang et al., 2020), followed by solving a reverse stochastic differential equation (SDE) associated with the forward process. Recent studies have rigorously established the convergence of diffusion models, demonstrating that the generated sample distribution approximates the target distribution (Lee et al., 2022; 2023; Chen et al., 2022; Benton et al., 2023; Chen et al., 2023; Li et al., 2024b; Gupta et al., 2024; Chen et al., 2024; Li et al., 2024a; Li & Yan, 2024; Li & Jiao, 2024; Li & Cai, 2024; Huang et al., 2024; Cai & Li, 2025; Li et al., 2025a).

As diffusion models become a dominant paradigm for generative modeling in domains such as image, video, and audio, the need for principled methods to modulate their output has grown significantly. For instance, when the data comprises multiple classes, one may seek to generate samples specific to a desired class. In practice, the standard approach is to use diffusion guidance (Dhariwal & Nichol, 2021; Ho & Salimans, 2021), a technique that enhances sample quality by incorporating an auxiliary conditional score function. This method combines the model's score estimate with the gradient of the log-probability of samples conditioned on the desired class through a weighted sum, enabling the generation of outputs with high perceptual quality when an appropriate guidance weight is applied. Reference (Karras et al., 2024) proposed to use a bad version of the model for guiding diffusion models.

### 1.1. Motivation

Despite the empirical success and widespread adoption of guidance methods, their theoretical foundations remain unexplored. A key question persists: why does guidance improve the quality of samples generated by diffusion models? Existing literature offers partial insights through case studies, analyzing guidance dynamics in limited scenarios such as mixtures of compactly supported distributions, isotropic Gaussian distributions, or linear guidance term (Chidambaram et al., 2024; Wu et al., 2024; Bradley &

---
[*]Equal contribution  [1]Department of Statistics, The Chinese University of Hong Kong, Hong Kong; Email: {genli,yuchenjiao}@cuhk.edu.hk. Correspondence to: Gen Li <genli@cuhk.edu.hk>.

*Proceedings of the 42$^{nd}$ International Conference on Machine Learning*, Vancouver, Canada, PMLR 267, 2025. Copyright 2025 by the author(s).

Nakkiran, 2024; Li et al., 2025b). However, the effect of guidance across general data distributions remains unknown, and we discover that the uniform improvement does not hold even for Gaussian mixture distributions (see Figure 1), which highlights a significant gap in our understanding.

## 1.2. Our Contributions

Motivated by the above discoveries, this paper investigates the improvement on the average of the reciprocal of classifier probabilities under general data distributions. We demonstrate that guidance preferentially enhances the generation of samples associated with higher classifier probabilities, which aligns with the primary motivation for adding guidance. Specifically, we prove that the expectation of the reciprocal of classifier probabilities decreases with guidance. This metric bears resemblance to the commonly used Inception Score (IS), a standard measure of sample quality (Salimans et al., 2016), which also considers the expectation of the (logarithmic) function of classifier probabilities. Furthermore, we extend our analysis to practical implementations, with discrete-errors and score estimation errors. We prove that the discrete-time processes approximate their continuous-time counterparts, ensuring the applicability of our theoretical results in practical settings.

**Comparison with prior works when restricted to specific distributions:** Existing works focus mainly on specific classes of distributions like GMMs, while our work provides a more general theoretical analysis. Here we compare our findings with prior works when restricted to specific distributions. In Wu et al. (2024), the authors demonstrate that $p_{c|X_0}(1|Y_1^w) \geq p_{c|X_0}(1|Y_1^0)$ holds under specific conditions, while we show that this inequality does not always hold. In addition, Chidambaram et al. (2024) argues that guidance can degrade the performance of diffusion models, as it may introduce mean overshoot and variance shrinkage. In contrast, our result shows that guidance can improve sample quality by generating more samples of high quality. Furthermore, Bradley & Nakkiran (2024) shows that classifier guidance can not generate samples from $p(x|c)^\gamma p(x)^{1-\gamma}$ for GMMs and establishes its connection to an alternative approach, i.e., the single-step predictor-corrector method, whose effectiveness in this specific setting remains unclear. In contrast, we directly analyze and demonstrate the effectiveness of CFG.

## 2. Background

In this section, we review basics about diffusion models, guidance, and their continuous limit. Throughout this paper, we shall use $n = 1, \cdots, N$ and $0 \leq t \leq 1$ to denote the discrete and continuous time steps, respectively.

### 2.1. Diffusion Models

Diffusion models are based on a forward process that progressively transforms data from a target distribution into a sequence of increasingly noisy representations. Starting from $X_0 \in \mathbb{R}^d$ drawn from the target distribution $p_{\text{data}}$, the forward process evolves as follows:

$$X_0 \sim p_{\text{data}}, \tag{1a}$$

$$X_n = \sqrt{1 - \beta_n} X_{n-1} + \sqrt{\beta_n} Z_n \quad n = 1, \cdots, N, \tag{1b}$$

where $0 < \beta_n < 1$ is the step-size, $\{Z_n\}_{1 \leq n \leq N} \overset{\text{i.i.d.}}{\sim} \mathcal{N}(0, I_d)$ is a sequence of independent Gaussian noise vectors. This process gradually converts the original distribution into standard Gaussian noise as $n$ increases.

An essential component of score-based diffusion models is the score function, defined as the gradient of the log-probability of the intermediate distributions in the forward process:

$$s_n^\star(x) := \nabla \log p_{X_n}(x), \quad 1 \leq n \leq N.$$

Assuming access to good approximations of the score functions, denoted $s_n(x) \approx s_n^\star(x)$, one can utilize them to reverse the forward process and generate samples resembling the target distribution. The reverse process is governed by:

$$Y_N \sim \mathcal{N}(0, I_d), \tag{2a}$$

$$Y_{n-1} = \frac{1}{\sqrt{1 - \beta_n}} \big(Y_n + \beta_n s_n(Y_n)\big) + \sqrt{\beta_n} Z_n, \tag{2b}$$

for $n = N, \cdots, 2$, where $Z_n \overset{\text{i.i.d.}}{\sim} \mathcal{N}(0, I_d)$ denotes another sequence of independent Gaussian noise vectors. This reverse process has been shown to gradually remove noise and guide the system back toward the target distribution, in the sense that the generated $Y_n$ has distribution close to that of $X_n$ in (1).

### 2.2. Guidance

Conditional diffusion models are designed to sample from the conditional distributions $p(\cdot|c)$, where $c$ represents a specific class label. This can be achieved by generalizing the unconditional diffusion model defined in (2), replacing $s_n(Y_n)$ with $s_n(Y_n|c)$, as shown below:

$$Y_N \sim \mathcal{N}(0, I_d), \tag{3a}$$

$$Y_{n-1} = \frac{1}{\sqrt{1 - \beta_n}} \big(Y_n + \beta_n s_n(Y_n \mid c)\big) + \sqrt{\beta_n} Z_n, \tag{3b}$$

for $n = N, \cdots, 2$, where $s_n(x|c)$ are good estimates of the gradient of the log-density function $p_{X_n \mid c}$, given the condition $c$. That is, $s_n(x|c) \approx s_n^\star(x|c) = \nabla \log p_{X_n \mid c}(x \mid c)$. The noise terms $Z_n \overset{\text{i.i.d.}}{\sim} \mathcal{N}(0, I_d)$ represent a sequence of independent Gaussian noise vectors.

To further enhance the quality of conditional sampling, researchers introduced guidance techniques. These methods aim to increase the posterior probability $p_{c\,|\,X_0}(c\,|\,Y_0)$ by modifying the reverse process as follows:

$$Y_{n-1}^w = \frac{1}{\sqrt{1-\beta_n}}\big(Y_n^w + \beta_n(s_n(Y_n^w\,|\,c)$$
$$+ w\nabla \log p_{c\,|\,X_n}(c\,|\,Y_n^w))\big) + \sqrt{\beta_n}Z_n, \quad (4)$$

where the guidance scale $w$ controls the strength of the modification. Furthermore, reverse process (4) can be approximated as

$$Y_{n-1}^w = \frac{1}{\sqrt{1-\beta_n}}\big(Y_n^w + \beta_n((1+w)s_n(Y_n^w\,|\,c)$$
$$- ws_n(Y_n^w)\big) + \sqrt{\beta_n}Z_n. \quad (5)$$

This approximation is derived from the observation that $\nabla \log p_{c\,|\,X_n}(c\,|\,x)) = s_n^\star(x\,|\,c) - s_n^\star(x)$, which is referred to as classifier free guidance (Ho & Salimans, 2021).

### 2.3. Continuous Time Limit

The discrete-time diffusion process described in Section 2.1 exhibits a natural correspondence to its continuous-time counterpart. Specifically, the forward process corresponds to the following stochastic differential equation (SDE):

$$dX_t = -\frac{1}{2(1-t)}X_t dt + \frac{1}{\sqrt{1-t}}dB_t, \quad (6a)$$
$$\text{with } X_0 \sim p_{\text{data}}, \quad \text{for } 0 \le t \le 1-\delta,$$

where $B_t$ denotes the standard Brownian motion, and $\delta > 0$ can be arbitrarily small. It transforms the data distribution into a standard Gaussian distribution as $t \to 1$. Similarly, the reverse process in (3) corresponds to the following continuous-time SDE:

$$dY_t = \Big(\frac{1}{2}Y_t + \nabla \log p_{X_{1-t}\,|\,c}(Y_t\,|\,c)\Big)\frac{dt}{t}$$
$$+ \frac{1}{\sqrt{t}}dB_t, \quad \text{for } \delta \le t \le 1.$$

This reverse SDE effectively transforms the noise distribution back toward the target distribution conditioned on $c$, guided by the conditional score function $\nabla \log p_{X_{1-t}\,|\,c}(Y_t\,|\,c)$. If the initialization $Y_\delta \sim p_{X_{1-\delta}\,|\,c}$, it is well-known that $Y_t$ has the same distribution with the reverse process of $X_t$, which is stated in the following lemma:

**Lemma 2.1.** *It can be shown that for $0 \le \tau \le t \le 1-\delta$,*

$$X_t\,|\,X_\tau \sim \mathcal{N}\Big(\sqrt{\frac{1-t}{1-\tau}}X_\tau, \frac{t-\tau}{1-\tau}I\Big), \quad (7)$$

*and if $Y_\delta \sim p_{X_{1-\delta}\,|\,c}$, then*

$$\{Y_t\} \stackrel{\mathrm{d}}{=} \{X_{1-t}\}, \quad \text{for } \delta \le t \le 1. \quad (8)$$

The above result can be found in Song et al. (2021b). When extending this framework to conditional sampling with guidance in (5), the reverse SDE becomes

$$dY_t^w = \Big(\frac{1}{2}Y_t^w + (1+w)\nabla \log p_{X_{1-t}\,|\,c}(Y_t^w\,|\,c)$$
$$- w\nabla \log p_{X_{1-t}}(Y_t^w)\Big)\frac{dt}{t} + \frac{1}{\sqrt{t}}dB_t. \quad (9)$$

The continuous-time framework provides a powerful perspective for understanding and analyzing score-based diffusion models.

## 3. Main Results

In this section, we shall present our main theorem and its proof. For the reverse process with guidance (9), we prove that after introducing a non-zero guidance into the diffusion process, the expectation of a specific decreasing function of the classifier probability will decrease as $t$ increases. This is formally stated in the following theorem.

**Theorem 3.1.** *Let*

$$\phi_t(y) := p_{c\,|\,X_{1-t}}(c\,|\,y)^{-1} \quad (10)$$

*which is a decreasing map of $p_{c\,|\,X_{1-t}}(c\,|\,y)$. It can be shown that for any $\delta < t < 1$,*

$$\phi_t(Y_t^w) - \mathbb{E}\big[\phi_{t+dt}(Y_{t+dt}^w)\,|\,Y_t^w\big] = \frac{w}{t}p_{c\,|\,X_{1-t}}(c\,|\,Y_t^w)^{-1}$$
$$\cdot \Big\|\nabla \log p_{X_{1-t}\,|\,c}(Y_t^w\,|\,c) - \nabla \log p_{X_{1-t}}(Y_t^w)\Big\|_2^2 dt, \quad (11)$$

*where $Y_t^w$ is defined in (9).*

The above result reveals that the average reciprocal of classifier probability $p_{c\,|\,X_{1-t}}(c\,|\,y)^{-1}$ decreases when we add non-zero guidance. When compared with the case without guidance, that is $w = 0$, the total expected improvement over the diffusion process is given by:

$$\int \frac{w}{t}p_{c\,|\,X_{1-t}}(c\,|\,Y_t^w)^{-1}$$
$$\cdot \Big\|\nabla \log p_{X_{1-t}\,|\,c}(Y_t^w\,|\,c) - \nabla \log p_{X_{1-t}}(Y_t^w)\Big\|_2^2 dt. \quad (12)$$

This result reflects an improvement in sample quality, as samples with higher classifier probabilities are favored.

The choice of $p_{c\,|\,X_{1-t}}(c\,|\,y)^{-1}$ in our analysis is primarily for technical considerations. It rewards more on the decrease of bad samples with small $p_{c\,|\,X_{1-t}}(c\,|\,y)$, which means it places greater emphasis on reducing the probability of generating low-quality or misclassified samples. This aligns with the initial motivation of introducing guidance. In practice, Inception Score (IS) is commonly employed to measure sample quality, which is related to the average

logarithm of the classifier probability $\mathbb{E}[\log p_{c\,|\,X_{1-t}}(c\,|\,y)]$. This is conceptually aligned with the metric in our analysis, with the difference being that IS adopts $\log p_{c\,|\,X_{1-t}}(c\,|\,y)$ as the weight, while we use $p_{c\,|\,X_{1-t}}(c\,|\,y)^{-1}$, but both aim to increase the ratio of high-quality samples (measured by the classifier probability). In addition, to address potential concerns, we note that although some practical limitations of IS have been identified (Barratt & Sharma, 2018), it remains a commonly used metric for evaluating sample quality in the study of diffusion guidance (Dhariwal & Nichol, 2021; Ho & Salimans, 2021). Moreover, in our theoretical analysis, we use the true conditional probability, which addresses the estimation issues discussed in Barratt & Sharma (2018).

Theorem 3.1 states that guidance improves the averaged reciprocal of the classifier probability rather than the classifier probability of each individual sample. This suggests that while guidance improves overall sample quality, it may lead to a decline in quality for a small subset of samples. This insight encourages the development of adaptive guidance methods that address this issue and achieve more uniform performance gains, which is a potential practical application of our theory.

Our main result is established through the following key observation, whose proof can be found in Section 4.1.

**Lemma 3.2.** *For any $\varepsilon > 0$ and $0 \le \tau \le t \le 1 - \varepsilon$, we have*

$$p_{c\,|\,X_t}(c\,|\,x)^{-1} = \mathbb{E}_{x_\tau \sim X_\tau}\big[p_{c\,|\,X_\tau}(c\,|\,x_\tau)^{-1}\,|\,X_t = x\big],$$
(13a)

*or equivalently, for any $\varepsilon \le \tau \le t \le 1$,*

$$p_{c\,|\,X_{1-\tau}}(c\,|\,y)^{-1} = \mathbb{E}_{y_t \sim Y_t}\big[p_{c\,|\,X_{1-t}}(c\,|\,y_t)^{-1}\,|\,Y_\tau = y\big],$$
(13b)

*where, $X_t$ and $Y_t$ are defined in* (6).

With Lemma 3.2 in hand, we are ready to prove our main theorem. Before diving into the proof details, we would like to first explain the main analysis idea: First, this result comes from the key observation that the function of reverse process, $p_{c|X_t}(c|X_t)^{-1}$, forms a martingale, as stated in Lemma 3.2, which is established through a careful decomposition of $p_{c|X_t}$ and $p_{X_\tau|X_t}$. Next, the guidance term $s_t(x|c) - s_t(x)$ in classifier-free guidance (CFG) aligns with the direction of $-\nabla p_{c|X_t}(c|x)^{-1} = p_{c|X_t}(c|x)^{-1}[s_t(x|c) - s_t(x)]$, which makes us expect that adding the guidance at time $t$ can decrease $\mathbb{E}_{x_\tau \sim X_\tau}\big[p_{c|X_\tau}(c|x_\tau)^{-1}|X_t = x\big]$ for all $\tau \le t$. Finally, to achieve the desired result, particular care must be taken in handling first- and second-order differential terms with respect to $t$ for the process $p_{c|X_{1-t}}(c|Y_t^w)^{-1}$ due to its randomness nature, which is completed in the following based on the technique of Ito's formula.

*Proof of Theorem 3.1.* The relation (13) in the above lemma gives us

$$
\begin{aligned}
0 = {} & \frac{1}{\delta}\Big\{\mathbb{E}\big[p_{c\,|\,X_{1-t-\delta}}(c\,|\,Y_{t+\delta})^{-1} \\
& \quad - p_{c\,|\,X_{1-t}}(c\,|\,Y_t)^{-1}\,|\,Y_t = y_t\big]\Big\} \\
= {} & \frac{\partial p_{c\,|\,X_{1-t}}(c\,|\,y)^{-1}}{\partial t}\Big|_{y=y_t} + \frac{1}{2t}\mathsf{Tr}\Big(\nabla^2 p_{c\,|\,X_{1-t}}(c\,|\,y_t)^{-1}\Big) \\
& + \nabla p_{c\,|\,X_{1-t}}(c\,|\,y_t)^{-1}\Big(\big(\tfrac{1}{2}y_t + \nabla \log p_{X_{1-t}\,|\,c}(y_t\,|\,c)\big)\tfrac{1}{t}\Big) \\
& + O(\delta),
\end{aligned}
$$
(14)

where the second relation is established in Section 4.3. Here, we let $\delta > 0$ be some small quantity, which depends only on $y_t, t$ and the property of $X_0$. Similarly, we have

$$
\begin{aligned}
& \frac{1}{\delta}\Big\{\mathbb{E}\big[p_{c\,|\,X_{1-t-\delta}}(c\,|\,Y_{t+\delta}^w)^{-1} \\
& \quad - p_{c\,|\,X_{1-t}}(c\,|\,Y_t^w)^{-1}\,|\,Y_t^w = y_t\big]\Big\} \\
= {} & \frac{\partial p_{c\,|\,X_{1-t}}(c\,|\,y)^{-1}}{\partial t}\Big|_{y=y_t} + \frac{1}{2t}\mathsf{Tr}\Big(\nabla^2 p_{c\,|\,X_{1-t}}(c\,|\,y_t)^{-1}\Big) \\
& + \nabla p_{c\,|\,X_{1-t}}(c\,|\,y_t)^{-1}\Big(\big(\tfrac{1}{2}y_t + (1+w)\nabla \log p_{X_{1-t}\,|\,c}(y_t\,|\,c) \\
& \quad - w\nabla \log p_{X_{1-t}}(y_t)\big)\tfrac{1}{t}\Big) + O(\delta).
\end{aligned}
$$
(15)

Comparing the above two relations leads to

$$
\begin{aligned}
& \mathbb{E}\big[\phi_{t+\delta}(Y_{t+\delta}^w)\,|\,Y_t^w\big] - \phi_t(Y_t^w) \\
= {} & \delta\frac{w}{t}\Big(\nabla \log p_{X_{1-t}\,|\,c}(Y_t^w\,|\,c) - \nabla \log p_{X_{1-\tau}}(Y_t^w)\Big) \\
& \cdot \nabla p_{c\,|\,X_{1-t}}(c\,|\,Y_t^w)^{-1} + O(\delta^2) \\
= {} & -\delta\frac{w}{t} p_{c\,|\,X_{1-t}}(c\,|\,Y_t^w)^{-1}\Big\|\nabla \log p_{X_{1-t}\,|\,c}(Y_t^w\,|\,c) \\
& \quad - \nabla \log p_{X_{1-t}}(Y_t^w)\Big\|_2^2 + O(\delta^2),
\end{aligned}
$$
(16)

where the second relation holds since

$$
\begin{aligned}
\nabla p_{c\,|\,X_{1-t}}(c\,|\,y)^{-1} = {} & -p_{c\,|\,X_{1-t}}(c\,|\,y)^{-1} \\
& \cdot \Big(\nabla \log p_{X_{1-t}\,|\,c}(y\,|\,c) - \nabla \log p_{X_{1-t}}(y)\Big).
\end{aligned}
$$
(17)

Then we can conclude the proof here.

$\square$

## 3.1. Numerical Validation

In this section, we present experimental results on the Gaussian Mixture Model (GMM) and ImageNet dataset to demonstrate that guidance does not uniformly enhance the quality of all samples. Instead, it improves overall sample quality by reducing the average reciprocal of the classifier probability. This observation empirically validate our theoretical findings.

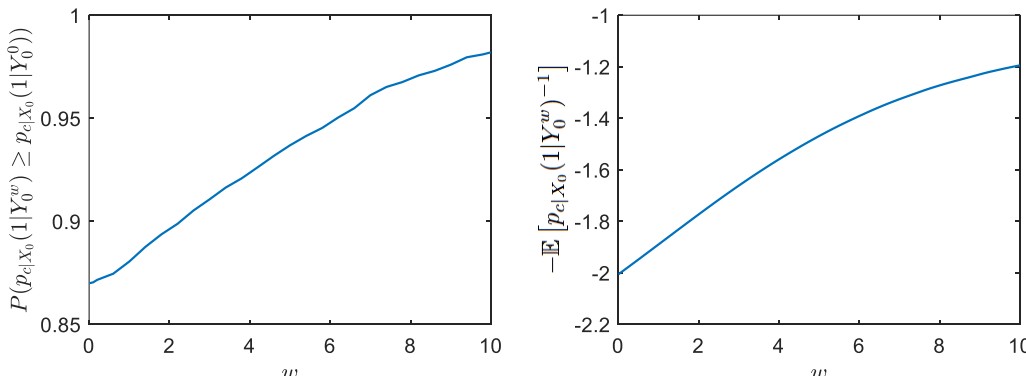

*Figure 1.* left: Ratio of samples with improved classifier probabilities for different guidance scales $w$; right: Expectation of $-p_{c \mid X_0}(1 \mid Y_0^w)^{-1}$ for varying $w$.

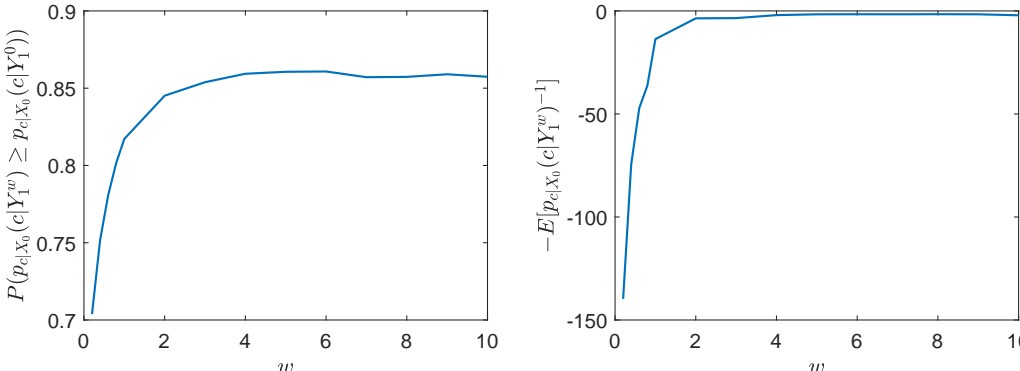

*Figure 2.* Experimental results on ImageNet dataset. left: Ratio of samples with improved classifier probabilities for different guidance scales $w$; right: Expectation of $-p_{c \mid X_0}(1 \mid Y_1^w)^{-1}$ for varying $w$.

**Gaussian Mixture Model:** Let us consider a distribution with two classes $c = 0, 1$, each with equal prior probability $p_c(0) = p_c(1) = 0.5$, in a one-dimensional data space ($d = 1$). The data distribution is defined as follows:

$$X_0 \mid c = 0 \ \sim \ \mathcal{N}(0,1)$$
$$X_0 \mid c = 1 \ \sim \ \frac{1}{2}\mathcal{N}(1,1) + \frac{1}{2}\mathcal{N}(-1,1).$$

According to the DDPM framework with guidance (5), the reverse process adopts the following update rule. Starting from $Y_N^w \sim \mathcal{N}(0,1)$, the process evolves for $n = N, \cdots, 2$:

$$Y_{n-1}^w = \frac{1}{\sqrt{\alpha_n}}\big(Y_n^w + (1 - \alpha_n)\big[ - w\nabla \log p_{X_{1-\overline{\alpha}_n}}(Y_n^w)$$
$$+ (1 + w)\nabla \log p_{X_{1-\overline{\alpha}_n} \mid c}(Y_n^w \mid c)\big]\big) + \sqrt{1 - \alpha_n}Z_n,$$
$$\tag{19}$$

where $Z_n \overset{\text{i.i.d.}}{\sim} \mathcal{N}(0,1)$ is a sequence of independent Gaussian random variables.

Here, we focus on the conditional class $c = 1$. The score functions $\nabla \log p_{X_{1-\overline{\alpha}_n} \mid c}(x \mid 1)$, $\nabla \log p_{X_{1-\overline{\alpha}_n}}(x)$, and the classifier probability $p_{c \mid X_{1-\overline{\alpha}_n}}(1 \mid x)$ are provided in Appendix B (cf. (44), (45), and (46)). To empirically validate our theoretical findings, we simulate the DDPM framework under different guidance scales $w$. Specifically, we fix $N = 4000$, vary $w$ from 0.01 to 10, and perform $10^4$ trials for each $w$. We compute $Y_1^w$ by implementing the reverse process in (19), and its counterpart $Y_1^0$ without guidance. For each trial, we evaluate classifier probability $p_{c \mid X_0}(1 \mid Y_1^w)$ and $p_{c \mid X_0}(1 \mid Y_1^0)$, and compute the empirical probability of $P(p_{c \mid X_0}(1 \mid Y_1^w) \geq p_{c \mid X_0}(1 \mid Y_1^0))$. In addition, we also calculate the average of $-p_{c \mid X_0}(1 \mid Y_1^w)^{-1}$ for various $w$. The results are shown in Figure 1.

**ImageNet dataset:** We conduct a numerical experiment on the ImageNet dataset. Specifically, we generate samples using a pre-trained diffusion model (Rombach et al., 2021) with varying values of the guidance

level $w$, and evaluate the classifier probabilities using the Inception v3 classifier (Szegedy et al., 2016). We compute two statistics: $P(p_{c\,|\,X_0}(1\,|\,Y_1^w) \geq p_{c\,|\,X_0}(1\,|\,Y_1^0))$ and $-\mathbb{E}[p_{c\,|\,X_0}(1\,|\,Y_1^w)^{-1}]$, averaged over 20000 random trials — 20 trials for each of the 1000 ImageNet categories. The experimental results are presented in Figure 2.

It is observed that the empirical probability $P(p_{c\,|\,X_0}(1\,|\,Y_1^w) \geq p_{c\,|\,X_0}(1\,|\,Y_1^0))$ is less than 1 for any $w < 10$, which indicates the guidance does not achieve uniform improvement in classifier probabilities. However, the average of $-p_{c\,|\,X_0}(1\,|\,Y_1^w)^{-1}$ increases with $w$, which explains why guidance effectively enhances sample quality, as predicted by Theorem 3.1. Moreover, we remark that the performance of diffusion models is commonly evaluated by two metrics in practice: diversity and sample quality. This study primarily focuses on the sample quality measured in a similar way as the Inception Score, which increases with $w$. However, prior work Ho & Salimans (2021) has demonstrated that large values of $w$ can significantly reduce sample diversity, leading to unsatisfactory performance in real-world applications.

### 3.2. Discretization and Robustness Analysis

Consider that practical algorithms operate in discrete time and are subject to score estimation errors, we provide a supplementary analysis of the discretization error and estimation error for completeness. Specifically, we aim to show the discrete-time process in (5) closely approximates the continuous-time process in (9), thereby validating the observation from Theorem 3.1 in practical settings. Since our primary focus is on the efficiency of diffusion guidance rather than establishing a convergence theory, the bounds and conditions derived here may not be tight.

In the following, we shall use $Y_t^{w,\text{cont}}$ to denote the continuous process of (9) in order to distinguish with (5), and let

$$\overline{\alpha}_n := \prod_{k=1}^{n} \alpha_k, \quad \text{with } \alpha_k := 1 - \beta_k \qquad (20)$$

satisfying

$$\overline{\alpha}_N = \frac{1}{N^{c_0}}, \qquad (21a)$$

$$\overline{\alpha}_{n-1} = \overline{\alpha}_n + \frac{c_1 \overline{\alpha}_n (1 - \overline{\alpha}_n) \log N}{N}, \qquad (21b)$$

where $c_0$ and $c_1$ are constants.

Before presenting the analysis result, we make the following assumptions. The first assumption states that faithful estimates of the score functions $s_n^\star(\cdot)$ and $s_n^\star(\cdot|c)$ are available for all intermediate steps $n$, as follows:

**Assumption 3.3.** We assume access to estimates $s_n(Y_{\overline{\alpha}_n}^{w,\text{cont}})$ and $s_n(Y_{\overline{\alpha}_n}^{w,\text{cont}}\,|\,c)$ for each $s_n^\star(Y_{\overline{\alpha}_n}^{w,\text{cont}})$ and

$s_n^\star(Y_{\overline{\alpha}_n}^{w,\text{cont}}\,|\,c)$ with the averaged $\ell_2$ score estimation error as

$$\frac{1}{N}\sum_{n=1}^{N} \mathbb{E}\Big[\big\|s_n(Y_{\overline{\alpha}_n}^{w,\text{cont}}\,|\,c)$$
$$- \nabla \log p_{X_{1-\overline{\alpha}_n}\,|\,c}(Y_{\overline{\alpha}_n}^{w,\text{cont}}\,|\,c)\big\|_2^2\Big] \leq \varepsilon_{\text{score}}^2; \quad (22a)$$

$$\frac{1}{N}\sum_{n=1}^{N} \mathbb{E}\Big[\big\|s_n(Y_{\overline{\alpha}_n}^{w,\text{cont}})$$
$$- \nabla \log p_{X_{1-\overline{\alpha}_n}}(Y_{\overline{\alpha}_n}^{w,\text{cont}})\big\|_2^2\Big] \leq \varepsilon_{\text{score}}^2. \quad (22b)$$

We further assume that the sample $Y_t^{w,\text{cont}}$, the score function $\nabla \log p_{X_{1-t}}(Y_t^{w,\text{cont}})$, and the conditional score function $\nabla \log p_{X_{1-t}\,|\,c}(Y_t^{w,\text{cont}}\,|\,c)$ have bounded second-order moment, which is stated in the following lemma.

**Assumption 3.4.** There exists some quantity $R$, such that the sum of the second-order moment of the following three random vectors are bounded by $R^2$, that is,

$$\mathbb{E}\Big[\big\|Y_t^{w,\text{cont}}\big\|_2^2 + \big\|\nabla \log p_{X_{1-t}}(Y_t^{w,\text{cont}})\big\|_2^2$$
$$+ \big\|\nabla \log p_{X_{1-t}\,|\,c}(Y_t^{w,\text{cont}}\,|\,c)\big\|_2^2\Big] \leq R^2. \qquad (23)$$

In addition, we consider the case with smooth score functions in this paper, which is stated below.

**Assumption 3.5.** Assume that $\nabla \log p_{X_t}(x)$ are Lipschitz for all $0 < t < 1$ such that

$$\big\|\nabla \log p_{X_t}(x_1) - \nabla \log p_{X_t}(x_2)\big\|_2 \leq L\|x_1 - x_2\|_2. \quad (24)$$

With the above assumptions, We could establish that the discrete-time process converges to the continuous-time process measured by the KL divergence. The proof is postponed to Appendix A.1.

**Theorem 3.6.** *Suppose that Assumptions 3.3, 3.4, and 3.5 hold true. Then the sampling process* (5) *with the learning rate schedule* (21) *satisfies*

$$\text{KL}(Y_{\overline{\alpha}_1}^{w,\text{cont}}, Y_1^w) \leq C\Big(\frac{(1+w^2)L^2 d \log^3 N}{N}$$
$$+ \frac{(1+w^4)L^2 R^2 \log^4 N}{N^2} + (1+w^2)\varepsilon_{\text{score}}^2 \log N\Big) \quad (25)$$

*for some constant $C > 0$ large enough, where $Y_{\overline{\alpha}_1}^{w,\text{cont}}$ and $Y_1^w$ are defined in* (9) *and* (5)*, respectively.*

This theorem proves that, after a sufficiently large number of iterations $N$, the sample distribution of the discrete-time process $Y_n^w$ converges to that of the continuous-time process $Y_{\overline{\alpha}_1}^{w,\text{cont}}$. The latter corresponds to data contaminated by noise with variance $1 - \overline{\alpha}_1$. According to Theorem 3.6, the

sampling process (5) with the learning rate schedule (21) satisfies

$$\mathbb{E}[p(c|Y_1^w)^{-1}] \leq \mathbb{E}[p(c|Y_{\overline{\alpha}_1}^{w,\text{cont}})^{-1}]$$
$$+ \mathbb{E}[(p(c|Y_1^w)^{-1} - 1)\mathbb{1}(p(c|Y_1^w)^{-1} > \tau)],$$

where $\tau$ is defined as the largest value satisfying

$$\mathsf{TV}(Y_{\overline{\alpha}_1}^{w,\text{cont}}, Y_1^w) \leq \mathbb{P}(p(c|Y_1^w)^{-1} > \tau).$$

This further implies the following relative influence from discretization, the ratio between the improvements of $Y_1^w$ and $Y_{\overline{\alpha}_1}^{w,\text{cont}}$ over $X_{\overline{\alpha}_1} = Y_{\overline{\alpha}_1}^{0,\text{cont}}$, obeys

$$\frac{\mathbb{E}[p(c|Y_{\overline{\alpha}_1}^{0,\text{cont}})^{-1}] - \mathbb{E}[p(c|Y_1^w)^{-1}]}{\mathbb{E}[p(c|Y_{\overline{\alpha}_1}^{0,\text{cont}})^{-1}] - \mathbb{E}[p(c|Y_{\overline{\alpha}_1}^{w,\text{cont}})^{-1}]}$$
$$\geq 1 - \frac{\mathbb{E}[(p(c|Y_1^w)^{-1} - 1)\mathbb{1}(p(c|Y_1^w)^{-1} > \tau)]}{\mathbb{E}[p(c|Y_{\overline{\alpha}_1}^{0,\text{cont}})^{-1}] - \mathbb{E}[p(c|Y_{\overline{\alpha}_1}^{w,\text{cont}})^{-1}]}. \quad (26)$$

Appendix A.2 presents numerical results for the relative error $\frac{\mathbb{E}[(p(c|Y_1^w)^{-1}-1)\mathbb{1}(p(c|Y_1^w)^{-1}>\tau)]}{\mathbb{E}[p(c|Y_{\overline{\alpha}_1}^{0,\text{cont}})^{-1}] - \mathbb{E}[p(c|Y_{\overline{\alpha}_1}^{w,\text{cont}})^{-1}]}$ evaluated under varying total variation distance thresholds $\tau$ on the ImageNet dataset.

# 4. Analysis

In this section, we shall provide details in the proof of main results.

## 4.1. Proof of Lemma 3.2

According to the equivalence between $X_t$ and $Y_t$ (see (8) in Lemma 2.1), it is sufficient to focus on the first relation. Recalling Lemma 2.1 again tells us

$$\mathbb{E}_{x_\tau \sim X_\tau}\left[p_{c|X_\tau}(c|x_\tau)^{-1} \mid X_t = x\right]$$
$$= \int_{x_\tau} p_{X_\tau|X_t,c}(x_\tau|x,c)p_{c|X_\tau}(c|x_\tau)^{-1}\mathrm{d}x_\tau$$
$$= \int_{x_\tau} \frac{p_{X_\tau|c}(x_\tau|c)(2\pi\sigma^2)^{-d/2}\exp(-\frac{\|x-\alpha x_\tau\|_2^2}{2\sigma^2})}{p_{X_t|c}(x|c)}$$
$$\cdot \frac{p_{X_\tau}(x_\tau)}{p_{X_\tau|c}(x_\tau|c)p_c(c)}\mathrm{d}x_\tau$$
$$= \frac{\int_{x_\tau} p_{X_\tau}(x_\tau)(2\pi\sigma^2)^{-d/2}\exp(-\frac{\|x-\alpha x_\tau\|_2^2}{2\sigma^2})\mathrm{d}x_\tau}{p_{X_t|c}(x|c)p_c(c)}$$
$$= \frac{p_{X_t}(x)}{p_{X_t|c}(x|c)p_c(c)} = p_{c|X_t}(c|x)^{-1},$$

where we let $\alpha = \sqrt{\frac{1-t}{1-\tau}}$ and $\sigma = \sqrt{\frac{t-\tau}{1-\tau}}$. Here, the first line is just the definition of conditional expectation; the second line comes from the Bayes rule and the relation (7); and the last line can be derived by applying the Bayes rule and the relation (7) again.

## 4.2. Preliminary Analysis of $p_{c|X_{1-t}}$

We begin by establishing some key properties of $p_{c|X_{1-t}}$ to support the proofs of our main results. Let $R < \infty$ be some quantity such that

$$\mathbb{P}(\|X_0\|_2 < R) > \frac{1}{2} \quad \text{and} \quad \mathbb{P}(\|X_0\|_2 < R \mid c) > \frac{1}{2}. \quad (27)$$

Then there exists some quantity $C_{t,k,R} > 0$ depending only on $t, k, R$, such that the following bounds hold:

$$\nabla^k p_{c|X_{1-t}}(c|y)^{-1} \leq \exp(C_{t,k,R}(1+\|y\|_2^2)); \quad (28a)$$
$$\frac{\partial^k p_{c|X_{1-t}}(c|y)^{-1}}{\partial t^k} \leq \exp(C_{t,k,R}(1+\|y\|_2^2)); \quad (28b)$$
$$\nabla^k \frac{\partial p_{c|X_{1-t}}(c|y)^{-1}}{\partial t} \leq \exp(C_{t,k,R}(1+\|y\|_2^2)), \quad (28c)$$

where $\nabla^k p_{c|X_{1-t}}(c|y)^{-1}$ denotes the $k$-th order gradient with respect to $y$ of function $p_{c|X_{1-t}}(c|y)^{-1}$.

In the following, we focus primarily on the gradient $\nabla p_{c|X_{1-t}}(c|y)^{-1}$, as the other bounds can be derived using similar techniques. Notice that $\nabla p_{c|X_{1-t}}(c|y)^{-1}$ satisfies the following decomposition:

$$\nabla p_{c|X_{1-t}}(c|y)^{-1}$$
$$= -p_{c|X_{1-t}}(c|y)^{-2}\nabla p_{c|X_{1-t}}(c|y)$$
$$= -p_{c|X_{1-t}}(c|y)^{-1}\nabla \log p_{c|X_{1-t}}(c|y)$$
$$= p_{c|X_{1-t}}(c|y)^{-1}\nabla\left[\log p_{X_{1-t}}(y) - \log p_{X_{1-t}|c}(y|c)\right]. \quad (29)$$

In addition, it can be shown later that

$$p_{c|X_{1-t}}(c|y)^{-1} \leq 2p_c(c)^{-1}\exp\left(\frac{(\|y\|_2 + \sqrt{t}R)^2}{2(1-t)}\right), \quad (30a)$$

and

$$\left\|\nabla \log p_{X_{1-t}}(y)\right\|_2 \lesssim \frac{\|y\|_2 + \sqrt{t}R}{1-t} + \frac{d}{\sqrt{1-t}}, \quad (30b)$$
$$\left\|\nabla \log p_{X_{1-t}|c}(y)\right\|_2 \lesssim \frac{\|y\|_2 + \sqrt{t}R}{1-t} + \frac{d}{\sqrt{1-t}}, \quad (30c)$$

where $f \lesssim g$ implies that there exists a universal constant $C > 0$ such that $f \leq Cg$. By inserting (30a) and (30b) into (29), the gradient $\nabla p_{c|X_{1-t}}(c|y)^{-1}$ can be controlled directly.

**Proof of Claim (30a) - (30c).** We begin with establishing (30a). First, according to Lemma 2.1, random variable $X_{1-t}|X_0$ follows Gaussian distribution $\mathcal{N}(\sqrt{t}X_0, (1-t)I)$.

Thus we have

$$p_{X_{1-t}}(y) = \int_{x_0} p_{X_0}(x_0) p_{X_{1-t}|X_0}(y|x_0) \mathrm{d}x_0$$

$$= \int_{x_0} p_{X_0}(x_0)(2\pi(1-t))^{-d/2} \exp\left(-\frac{\|y-\sqrt{t}x_0\|_2^2}{2(1-t)}\right) \mathrm{d}x_0$$

$$\leq (2\pi(1-t))^{-d/2} \int_{x_0} p_{X_0}(x_0) \mathrm{d}x_0$$

$$= (2\pi(1-t))^{-d/2}. \tag{31}$$

Moreover, recalling the definition of $R$ in (27), we have

$$p_{X_{1-t}|c}(y|c) \geq p_{X_{1-t}, \|X_0\|_2 < R|c}(y|c)$$

$$= \mathbb{P}(\|X_0\|_2 < R|c) p_{X_{1-t}|c, \|X_0\|_2 < R}(y|c, \|X_0\|_2 < R)$$

$$\geq \frac{1}{2} \inf_{x_0: \|x_0\|_2 < R} (2\pi(1-t))^{-d/2} \exp\left(-\frac{\|y-\sqrt{t}x_0\|_2^2}{2(1-t)}\right) \tag{32}$$

$$\geq \frac{1}{2}(2\pi(1-t))^{-d/2} \exp\left(-\frac{(\|y\|_2 + \sqrt{t}R)^2}{2(1-t)}\right), \tag{33}$$

where $p_{X_{1-t}, \|X_0\|_2 < R|c}(y|c)$ denotes the joint probability density of $X_{1-t}$ and the binary random variable indicating $\|X_0\|_2 < R$ or not, and $p_{X_{1-t}|c, \|X_0\|_2 < R}(y|c)$ denotes the probability density of $X_{1-t}$ conditioned on the class label $c$ and $\|X_0\|_2 < R$. Combining (31) and (33), we have

$$p_{c|X_{1-t}}(c|y)^{-1} = \frac{p_{X_{1-t}}(y)}{p_c(c) p_{X_{1-t}|c}(y|c)}$$

$$\leq 2 p_c(c)^{-1} \exp\left(\frac{(\|y\|_2 + \sqrt{t}R)^2}{2(1-t)}\right).$$

Next, we shall prove (30b). For $t < 1$, recalling that the random variable $X_{1-t}|X_0$ follows Gaussian distribution $\mathcal{N}(\sqrt{t}X_0, (1-t)I)$, the score function has the following expression

$$\nabla \log p_{X_{1-t}}(y)$$

$$= -p_{X_{1-t}}(y)^{-1} \int_{x_0} p_{X_0}(x_0)(2\pi(1-t))^{-d/2}$$

$$\cdot \exp\left(-\frac{\|y-\sqrt{t}x_0\|_2^2}{2(1-t)}\right) \frac{y-\sqrt{t}x_0}{1-t} \mathrm{d}x_0$$

$$= -\int_{x_0} p_{X_0|X_{1-t}}(x_0|y) \frac{y-\sqrt{t}x_0}{1-t} \mathrm{d}x_0. \tag{34}$$

Moreover, noticing that for any $D > 0$,

$$\left\|\nabla \log p_{X_{1-t}}(y)\right\|_2$$

$$= \int_{x_0: \left\|\frac{y-\sqrt{t}x_0}{\sqrt{1-t}}\right\|_2 \leq D} p_{X_0|X_{1-t}}(x_0|y) \left\|\frac{y-\sqrt{t}x_0}{1-t}\right\|_2 \mathrm{d}x_0$$

$$+ \left\|p_{X_{1-t}}(y)^{-1} \int_{x_0: \left\|\frac{y-\sqrt{t}x_0}{\sqrt{1-t}}\right\|_2 > D} p_{X_0}(x_0)(2\pi(1-t))^{-d/2} \right.$$

$$\left. \cdot \exp\left(-\frac{\|y-\sqrt{t}x_0\|_2^2}{2(1-t)}\right) \frac{y-\sqrt{t}x_0}{1-t} \mathrm{d}x_0 \right\|_2.$$

For the first term, we have

$$\int_{\frac{\|y-\sqrt{t}x_0\|}{\sqrt{1-t}} \leq D} p_{X_0|X_{1-t}}(x_0|y) \frac{\|y-\sqrt{t}x_0\|}{1-t} \mathrm{d}x_0 \leq \frac{D}{\sqrt{1-t}}.$$

For the second term, noticing that

$$p_{X_{1-t}}(y) \geq \frac{1}{2}(2\pi(1-t))^{-d/2} \exp\left(-\frac{(\|y\|_2 + \sqrt{t}R)^2}{2(1-t)}\right),$$

we have

$$\left\|p_{X_{1-t}}(y)^{-1} \int_{x_0: \left\|\frac{y-\sqrt{t}x_0}{\sqrt{1-t}}\right\|_2 > D} p_{X_0}(x_0)(2\pi(1-t))^{-\frac{d}{2}} \right.$$

$$\left. \cdot \exp\left(-\frac{\|y-\sqrt{t}x_0\|_2^2}{2(1-t)}\right) \frac{y-\sqrt{t}x_0}{1-t} \mathrm{d}x_0 \right\|_2$$

$$\leq 2 \exp\left(\frac{(\|y\|_2 + \sqrt{t}R)^2}{2(1-t)}\right) \int_{x_0: \left\|\frac{y-\sqrt{t}x_0}{\sqrt{1-t}}\right\|_2 > D} p_{X_0}(x_0)$$

$$\cdot \exp\left(-\frac{\|y-\sqrt{t}x_0\|_2^2}{2(1-t)}\right) \left\|\frac{y-\sqrt{t}x_0}{1-t}\right\|_2 \mathrm{d}x_0$$

$$\lesssim \frac{2}{\sqrt{1-t}} \exp\left(\frac{(\|y\|_2 + \sqrt{t}R)^2}{2(1-t)} - cD^2 + cd\right),$$

where $c$ is a universal constant.

By choosing

$$D = C\left(\frac{\|y\|_2 + \sqrt{t}R}{\sqrt{1-t}} + d\right)$$

for some constant $C > 0$ large enough, we have

$$\left\|\nabla \log p_{X_{1-t}}(y)\right\|_2 \leq \frac{2D}{\sqrt{1-t}} \lesssim \frac{\|y\|_2 + \sqrt{t}R}{1-t} + \frac{d}{\sqrt{1-t}}.$$

Similarly, we could derive that

$$\left\|\nabla \log p_{X_{1-t}|c}(y|c)\right\|_2 \lesssim \frac{\|y\|_2 + \sqrt{t}R}{1-t} + \frac{d}{\sqrt{1-t}}.$$

### 4.3. Proof of Claim (14)

We provide a detailed proof of Claim (14) by analyzing the decomposition of the expectation. We start by decomposing the expectation as follows:

$$\mathbb{E}\left[p_{c|X_{1-t-\delta}}(c|Y_{t+\delta})^{-1} - p_{c|X_{1-t}}(c|Y_t)^{-1} \mid Y_t = y_t\right]$$

$$= \mathbb{E}\left[p_{c|X_{1-t}}(c|Y_{t+\delta})^{-1} - p_{c|X_{1-t}}(c|Y_t)^{-1} \mid Y_t = y_t\right]$$

$$+ \mathbb{E}\left[p_{c|X_{1-t-\delta}}(c|Y_{t+\delta})^{-1} - p_{c|X_{1-t}}(c|Y_{t+\delta})^{-1} \mid Y_t = y_t\right]$$

In the following, we shall analyze these two terms separately.

**Analysis of the first term.** Applying Ito's formula gives us

$$p_{c\,|\,X_{1-t}}(c\,|\,Y_{t+\delta})^{-1} - p_{c\,|\,X_{1-t}}(c\,|\,Y_t)^{-1}$$

$$= \int_t^{t+\delta} \left\{ \frac{1}{2s}\mathsf{Tr}\Big(\nabla^2 p_{c\,|\,X_{1-t}}(c\,|\,Y_s)^{-1}\Big)\mathrm{d}s \right.$$

$$+ \nabla p_{c\,|\,X_{1-t}}(c\,|\,Y_s)^{-1} \cdot \left(\left(\frac{1}{2}Y_s \right.\right.$$

$$\left.\left.\left. + \nabla\log p_{X_{1-s}\,|\,c}(Y_s\,|\,c)\right)\frac{\mathrm{d}s}{s} + \frac{1}{\sqrt{s}}\mathrm{d}B_s\right)\right\}. \quad (35)$$

We further decompose the first term by using Ito's formula again as

$$\mathsf{Tr}\Big(\nabla^2 p_{c\,|\,X_{1-t}}(c\,|\,Y_s)^{-1}\Big) - \mathsf{Tr}\Big(\nabla^2 p_{c\,|\,X_{1-t}}(c\,|\,Y_t)^{-1}\Big)$$

$$= \int_t^s \left\{ \frac{1}{2r}\mathsf{Tr}\Big(\nabla^2\mathsf{Tr}\Big(\nabla^2 p_{c\,|\,X_{1-t}}(c\,|\,Y_r)^{-1}\Big)\Big)\mathrm{d}r \right.$$

$$+ \nabla\mathsf{Tr}\Big(\nabla^2 p_{c\,|\,X_{1-t}}(c\,|\,Y_r)^{-1}\Big) \cdot \left(\left(\frac{1}{2}Y_r \right.\right.$$

$$\left.\left.\left. + \nabla\log p_{X_{1-r}\,|\,c}(Y_r\,|\,c)\right)\frac{\mathrm{d}r}{r} + \frac{1}{\sqrt{r}}\mathrm{d}B_r\right)\right\}. \quad (36)$$

According to bound (28a), we have

$$\mathbb{E}\Big[\mathsf{Tr}\Big(\nabla^2\mathsf{Tr}\Big(\nabla^2 p_{c\,|\,X_{1-t}}(c\,|\,Y_r)^{-1}\Big)\Big)\,|\,Y_t = y_t\Big]$$

$$\le \mathbb{E}\Big[\exp(C_{r,4,R} + C_{r,4,R}\|Y_r\|_2^2)\,|\,Y_t = y_t\Big] < \infty \quad (37)$$

and

$$\mathbb{E}\Big[\nabla\mathsf{Tr}\Big(\nabla^2 p_{c\,|\,X_{1-t}}(c\,|\,Y_r)^{-1}\Big) \cdot \left(\left(\frac{1}{2}Y_r \right.\right.$$

$$\left.\left. + \nabla\log p_{X_{1-r}\,|\,c}(Y_r\,|\,c)\right)\,|\,Y_t = y_t\Big] < \infty. \quad (38)$$

Inserting (37) and (38) into (36), we have for $t \le s \le t+\delta$,

$$\mathsf{Tr}\Big(\nabla^2 p_{c\,|\,X_{1-t}}(c\,|\,Y_s)^{-1}\Big)$$

$$= \mathsf{Tr}\Big(\nabla^2 p_{c\,|\,X_{1-t}}(c\,|\,Y_t)^{-1}\Big) + O(\delta). \quad (39)$$

Similarly, we could get that for $t \le s \le t+\delta$,

$$\mathbb{E}\Big[\nabla p_{c\,|\,X_{1-t}}(c\,|\,Y_s)^{-1} \cdot \left(\left(\frac{1}{2}Y_s \right.\right.$$

$$\left.\left. + \nabla\log p_{X_{1-s}\,|\,c}(Y_s\,|\,c)\right)\,|\,Y_t = y_t\Big]$$

$$= \nabla p_{c\,|\,X_{1-t}}(c\,|\,y_t)^{-1} \cdot \left(\left(\frac{1}{2}y_t \right.\right.$$

$$\left.\left. + \nabla\log p_{X_{1-t}\,|\,c}(y_t\,|\,c)\right) + O(\delta). \quad (40)$$

Inserting (39) and (40) into (35), we have

$$\frac{1}{\delta}\mathbb{E}\left[p_{c\,|\,X_{1-t}}(c\,|\,Y_{t+\delta})^{-1} - p_{c\,|\,X_{1-t}}(c\,|\,Y_t)^{-1}|Y_t = y_t\right]$$

$$= \frac{1}{2t}\mathsf{Tr}\Big(\nabla^2 p_{c\,|\,X_{1-t}}(c\,|\,y_t)^{-1}\Big) + \nabla p_{c\,|\,X_{1-t}}(c\,|\,y_t)^{-1}$$

$$\cdot \left(\left(\frac{1}{2}y_t + \nabla\log p_{X_{1-t}\,|\,c}(y_t\,|\,c)\right)\frac{1}{t} + O(\delta).$$

**Analysis of the second term.** The second term can be expressed as:

$$\mathbb{E}\Big[p_{c\,|\,X_{1-t-\delta}}(c\,|\,Y_{t+\delta})^{-1} - p_{c\,|\,X_{1-t}}(c\,|\,Y_{t+\delta})^{-1}\,|\,Y_t = y_t\Big]$$

$$= \mathbb{E}\left[\int_t^{t+\delta}\frac{\partial}{\partial s}p_{c\,|\,X_{1-s}}(c\,|\,Y_{t+\delta})^{-1}\mathrm{d}s\,|\,Y_t = y_t\right].$$

Similar to the analysis of the first term, we notice that

$$\frac{\partial}{\partial s}p_{c\,|\,X_{1-s}}(c\,|\,y)^{-1} - \frac{\partial}{\partial t}p_{c\,|\,X_{1-t}}(c\,|\,y)^{-1}$$

$$= \int_t^s \frac{\partial^2}{\partial r^2}p_{c\,|\,X_{1-r}}(c\,|\,y)^{-1}\mathrm{d}r,$$

and according to (28b),

$$\mathbb{E}\left[\frac{\partial^2}{\partial r^2}p_{c\,|\,X_{1-r}}(c\,|\,Y_{t+\delta})^{-1}\,|\,Y_t = y_t\right] < \infty.$$

Thus we have

$$\frac{1}{\delta}\mathbb{E}\Big[p_{c\,|\,X_{1-t-\delta}}(c\,|\,Y_{t+\delta})^{-1} - p_{c\,|\,X_{1-t}}(c\,|\,Y_{t+\delta})^{-1}\,|\,Y_t = y_t\Big]$$

$$= \frac{\partial}{\partial t}p_{c\,|\,X_{1-t}}(c\,|\,y_t)^{-1} + O(\delta).$$

Combining the above two relations, we could get our desired result.

## 5. Discussion

In this paper, we present a theoretical analysis of the impact of guidance in diffusion models under general data distributions. Specifically, we demonstrate that guidance in the continuous-time process can enhance the sampling process by generating more high-quality samples — those associated with higher classifier probabilities — in the average sense. Additionally, we prove that the practical discrete-time process converges to the above analyzed continuous-time process, as the number of iterations goes to infinity. These results provide a theoretical foundation for the empirical success of guidance methods.

In this paper, the convergence analysis in Theorem 3.6 is included primarily for completeness. The dependencies on $d$, $L$ and $\varepsilon$ may not be optimal, and the smoothness condition might not be necessary. Future research could focus on establishing tighter bounds or analyzing under more general bounds, to broaden the applicability and improve the convergence rate. In addition, we are interested in extending these results to the concept of Inception Score (IS), demonstrating similar findings when the weights used in IS are applied.

## Acknowledgements

Gen Li is supported in part by the Chinese University of Hong Kong Direct Grant for Research and the Hong Kong Research Grants Council ECS 2191363.

## Impact Statement

This paper presents work whose goal is to advance the field of Machine Learning. There are many potential societal consequences of our work, none which we feel must be specifically highlighted here.

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

# A. Discretization and Robust Analysis

## A.1. Proof of Theorem 3.6

Here, we provide a brief sketch for this result. With similar analysis as Chen et al. (2022, Section 5),

$$
\begin{aligned}
&\mathsf{KL}(Y^{w,\mathsf{cont}}_{\overline{\alpha}_1}, Y^w_1) \\
&\leq \sum_{n=2}^{N} \mathbb{E} \int_{\overline{\alpha}_n}^{\overline{\alpha}_{n-1}} \big\| (1+w)[s_n(Y^{w,\mathsf{cont}}_{\overline{\alpha}_n} \,|\, c) - \nabla \log p_{X_{1-t}\,|\,c}(Y^{w,\mathsf{cont}}_t \,|\, c)] \\
&\qquad\qquad - w[s_n(Y^{w,\mathsf{cont}}_{\overline{\alpha}_n}) - \nabla \log p_{X_{1-t}}(Y^{w,\mathsf{cont}}_t)] \big\|_2^2 \frac{\mathrm{d}t}{t} + \mathsf{KL}(Y^{w,\mathsf{cont}}_{\overline{\alpha}_N}, Y^w_N).
\end{aligned}
\tag{41}
$$

Then it can be shown that

$$
\begin{aligned}
&\mathbb{E} \int_{\overline{\alpha}_n}^{\overline{\alpha}_{n-1}} \big\| s_n^\star(Y^{w,\mathsf{cont}}_{\overline{\alpha}_n}) - \nabla \log p_{X_{1-t}}(Y^{w,\mathsf{cont}}_t) \big\|_2^2 \frac{\mathrm{d}t}{t} \\
&\leq L^2 \mathbb{E} \int_{\overline{\alpha}_n}^{\overline{\alpha}_{n-1}} \big\| Y^{w,\mathsf{cont}}_{\overline{\alpha}_n} - Y^{w,\mathsf{cont}}_t \big\|_2^2 \frac{\mathrm{d}t}{t} \\
&\leq L^2 \mathbb{E} \int_{\overline{\alpha}_n}^{\overline{\alpha}_{n-1}} \bigg\| \int_{\overline{\alpha}_n}^{t} \bigg\{ \Big( \frac{Y^{w,\mathsf{cont}}_\tau}{2} + (1+w)\nabla \log p_{X_{1-\tau}\,|\,c}(Y^{w,\mathsf{cont}}_\tau \,|\, c) - w \nabla \log p_{X_{1-\tau}}(Y^{w,\mathsf{cont}}_\tau) \Big) \frac{\mathrm{d}\tau}{\tau} + \frac{\mathrm{d}B_\tau}{\sqrt{\tau}} \bigg\} \bigg\|_2^2 \frac{\mathrm{d}t}{t} \\
&\lesssim L^2((1+w)^2 R^2(1-\alpha_n) + d)(1-\alpha_n)^2.
\end{aligned}
$$

Inserting the above relation, Assumption 3.3, and Assumption 3.4 into (41) leads to our desired result.

## A.2. Numerical Validation

For different values of $\mathsf{TV}(Y^{w,\mathsf{cont}}_{\overline{\alpha}_1}, Y^w_1)$, we empirically validate the aforementioned result on the ImageNet dataset. Specifically, we generate $2 \times 10^4$ samples $Y^w_1$ under various guidance level $w$ and their counterparts $Y^w_0$ without guidance by using a pre-trained diffusion model (Rombach et al., 2021), and evaluate the classifier probability $p(c|Y^w_1)$ and $p(c|Y^0_1)$ by using the Inception v3 classifier (Szegedy et al., 2016). Finally, we evaluate the relative error in (26). Here we use $\mathbb{E}[p(c|Y^0_1)^{-1}] - \mathbb{E}[p(c|Y^w_1)^{-1}]$ as an estimate of $\mathbb{E}[p(c|Y^{0,\mathsf{cont}}_{\overline{\alpha}_1})^{-1}] - \mathbb{E}[p(c|Y^{w,\mathsf{cont}}_{\overline{\alpha}_1})^{-1}]$, and calculate the ratio of empirical average

$$
\frac{\mathbb{E}[(p(c|Y^w_1)^{-1} - 1)\mathbb{1}(p(c|Y^w_1)^{-1} > \tau)]}{\mathbb{E}[p(c|Y^0_1)^{-1}] - \mathbb{E}[p(c|Y^w_1)^{-1}]}.
$$

The results are presented in the following table for various values of the TV distance and $w$, which indicate that the relative error remains small, particularly for practical choices of $w \geq 1$.

*Table 1.* Empirical values of $\frac{\mathbb{E}[(p(c|Y^w_1)^{-1} - 1)\mathbb{1}(p(c|Y^w_1)^{-1} > \tau)]}{\mathbb{E}[p(c|Y^0_1)^{-1}] - \mathbb{E}[p(c|Y^w_1)^{-1}]}$ under different $w$ and TV.

| TV | $w = 0.2$ | 0.4 | 0.6 | 0.8 | 1 | 2 | 3 | 4 |
|---|---|---|---|---|---|---|---|---|
| 0.30 | 0.447 | 0.196 | 0.115 | 0.085 | 0.029 | 0.006 | 0.006 | 0.002 |
| 0.10 | 0.440 | 0.194 | 0.114 | 0.085 | 0.029 | 0.006 | 0.005 | 0.002 |

# B. Basis Calculations of GMM

Consider a GMM defined as:

$$
X_0 \sim \sum_{k=1}^{K} \pi_k \mathcal{N}(\mu_k, 1),
\tag{42}
$$

where $\pi_k$ is the mixing coefficient of the $k$-th component, and $\mu_k$ is its mean. By Lemma 2.1, we have

$$X_{1-\overline{\alpha}_n} \sim \sum_{k=1}^{K} \pi_k \mathcal{N}\big(\sqrt{\overline{\alpha}_n}\mu_k, 1\big)$$

$$p_{X_{1-\overline{\alpha}_n}}(x) = \sum_{k=1}^{K} \pi_k (2\pi)^{-1/2} \exp\left(-\frac{(x - \sqrt{\overline{\alpha}_n}\mu_k)^2}{2}\right).$$

The gradient of the log-density $\log p_{X1-\overline{\alpha}_n}(x)$ can be computed as:

$$\nabla \log p_{X_{1-\overline{\alpha}_n}}(x) = \frac{\nabla p_{X_{1-\overline{\alpha}_n}}(x)}{p_{X_{1-\overline{\alpha}_n}}(x)} = -\sum_{k=1}^{K} \pi_k^n \big(x - \sqrt{\overline{\alpha}_n}\mu_k\big) = -x + \sqrt{\overline{\alpha}_n}\sum_{k=1}^{K} \pi_k^n \mu_k, \tag{43}$$

where

$$\pi_k^n = \frac{\pi_k \exp\left(-\frac{(x-\sqrt{\overline{\alpha}_n}\mu_k)^2}{2}\right)}{\sum_{i=1}^{K} \pi_i \exp\left(-\frac{(x-\sqrt{\overline{\alpha}_n}\mu_i)^2}{2}\right)}.$$

Using this setup for specific cases $(K = 2, 3)$ leads to

$$\nabla \log p_{X_{1-\overline{\alpha}_n} \mid c}(x \mid 1) = -x + \frac{\sqrt{\overline{\alpha}_n}(1 - \exp(-2\sqrt{\overline{\alpha}_n}x))}{1 + \exp(-2\sqrt{\overline{\alpha}_n}x)}; \tag{44}$$

$$\nabla \log p_{X_{1-\overline{\alpha}_n}}(x) = -x + \frac{\sqrt{\overline{\alpha}_n}(1 - \exp(-2\sqrt{\overline{\alpha}_n}x))}{1 + \exp(-2\sqrt{\overline{\alpha}_n}x) + 2\exp\left(\frac{\overline{\alpha}_n}{2} - \sqrt{\overline{\alpha}_n}x\right)}. \tag{45}$$

Additionally, the classifier probability $p_{c \mid X_{1-\overline{\alpha}_n}}(1 \mid x)$ is given by

$$p_{c \mid X_{1-\overline{\alpha}_n}}(1 \mid x) = \frac{p_{X_{1-\overline{\alpha}_n} \mid c}(x \mid c)p(c)}{p_{X_{1-\overline{\alpha}_n}}(x)} = \frac{1 + \exp(-2\sqrt{\overline{\alpha}_n}x)}{1 + \exp(-2\sqrt{\overline{\alpha}_n}x) + 2\exp\left(\frac{\overline{\alpha}_n}{2} - \sqrt{\overline{\alpha}_n}x\right)}. \tag{46}$$

