# OpenReview forum: "Provable Efficiency of Guidance in Diffusion Models for General Data Distribution"
_ICML.cc/2025/Conference — ICML 2025 poster_

### Official Review · Reviewer_qwwj · 2025-03-07

**Overall Recommendation:** 1

**Summary:**

This paper presents a theoretical analysis of classifier-free guidance (CFG) in diffusion models, demonstrating that guidance enhances sample quality by reducing the expected ratio of poor samples, as measured by classifier probability. The authors establish a connection between their proposed metric and the Inception Score, a widely used evaluation metric for diffusion model sample quality, thereby justifying the choice of their metric. To validate their theoretical findings, the authors provide a one-dimensional experiment involving Gaussian Mixtures, which serves as a proof-of-concept for their approach.

**Claims And Evidence:**

The main result, Theorem 3.1, appears to be theoretically sound and relevant to the current literature on classifier-free guidance (CFG). However, I am concerned that the paper lacks sufficient empirical evidence to support this theoretical contribution. The experimental section is unfortunately very weak, which undermines the overall impact of the paper. To strengthen the manuscript, I would recommend that the authors invest significant effort into improving the experimental evaluation, including more comprehensive and rigorous experiments that demonstrate the practical effectiveness of their approach. Furthermore, giving more detailed insight and explanation regarding the main theorem and its contribution would benefit for the future readers. In its current form, I do not believe that the paper meets the standards expected for an ICML conference paper.

**Essential References Not Discussed:**

It is disappointing that the authors have not cited any related work, especially given the substantial body of research currently focused on explaining classifier-free guidance (CFG) from a theoretical perspective. For instance, relevant studies can be found in the following papers: https://arxiv.org/abs/2409.13074, https://arxiv.org/abs/2403.01639, and https://arxiv.org/abs/2408.09000. Including references to these works would provide valuable context and strengthen the paper's contribution to the field.

**Experimental Designs Or Analyses:**

There is only one experiment in one dimension. As mentioned above, even this experiment and its results do not seem completely valid.

**Methods And Evaluation Criteria:**

Unfortunately, the paper falls short in its experimental evaluation. The authors only present a single experiment on a one-dimensional Gaussian Mixture, which is insufficient to demonstrate the effectiveness of their approach. To strengthen the paper, I would recommend that the authors conduct additional experiments on higher-dimensional models, including Gaussian Mixtures, as well as real-world datasets. This would provide a more comprehensive understanding of the proposed method's performance and limitations.

Furthermore, I have concerns regarding the example presented on page 8. The authors claim that P(p_{c|X_0}(1|Y_0^w) \geq p_{c|X_0}(1|Y_0^0)) is less than one, but the curve suggests that this may not be the case for sufficiently large values of w. Moreover, this specific scenario has been previously analyzed in "What does guidance do? A fine-grained analysis in a simple setting" by Chidambaram et al., who demonstrated that classifier-free guidance (CFG) in one dimension can lead to flawed results, including mean overshoot and variance shrinkage. In light of these findings, it is essential that the authors perform experiments in larger dimensions and on real-world datasets to validate their claims and demonstrate the robustness of their approach.

**Other Comments Or Suggestions:**

The authors justify their chosen metric by demonstrating its relationship to the Inception Score. However, it is important to note that the Inception Score is considered outdated and is no longer widely used in recent work on diffusion models. For instance, a well-known study in the field highlights the shortcomings of this metric https://arxiv.org/pdf/1801.01973. I strongly recommend that the authors address this issue by discussing the relevance of their metric in light of these criticisms and by relating the findings of their paper to the proposed metric. This would help clarify the metric's applicability and strengthen the paper's contribution.

There is a typo on end of page 2, where you have double brackets )) and should have just one: ∇ log pc | Xn (c | x)).
Your title is "Title Suppressed Due to Excessive Size" so please make sure to amend this accordingly.
Page 2 also mentions multiple times that Z_n represents i.i.d. Gaussian noise, this could be avoided for conciseness and clarity.

**Other Strengths And Weaknesses:**

I would also strongly advise the authors to move all the proofs to the appendix and to only highlight the main results in the main paper, as well as add related work and stronger experimental results.

**Questions For Authors:**

No questions.

**Relation To Broader Scientific Literature:**

There is currently an increasing amount of theoretical works that are attempting to explain CFG, so this paper is timely and relevant.

**Theoretical Claims:**

I did not find an error in the calculations in the paper.

---

> ### Author Rebuttal · Authors · 2025-03-31
>
> Thanks for your valuable questions. Below we provide a detailed point-by-point response.
>
> **Experiments on real dataset.** We have added experiments on the ImageNet dataset to validate our theory. Please refer to our response to Reviewer EKNn of **Experiments on real dataset**.
>
> **Further explanations of the main theorem.**
> We agree that a more detailed explanation of Theorem 3.1 would enhance clarity. In the revised version, we will add the explanation of main analysis idea (see ''Explanation of main analysis idea'' in our response to Reviewer 5U8H), detailed comparison with prior theory (see ''Comparison with prior works'' in this response), explanation of IS (see ''Issue of Inception Score'' in this reponse), and experiments on Imagenet dataset (see ''Experiments on real dataset'' in our response to Reviewer EKNn) to address this comment.
>
> **Clarification of the toy example.**
> Originally, we state that *''$P(p_{1|X_0}(1|Y_0^w)\ge p_{1|X_0}(1 | Y_0^0))$ is less than $1$, which indicates the guidance may not achieve uniform improvement in classifier probabilities''* for the toy example. To make it more accurate, we will revise it to *''$P(p_{1 | X_0}(1| Y_0^w)\ge p_{1| X_0}(1| Y_0^0)) < 1$ for any $w < 10$, which indicates ...''*, since we have numerically confirmed that this holds true for all $w < 10$, which covers the practical range of $w$ in typical applications.
>
> **Relation to prior works on toy examples.** We agree with the existence of prior analyses of classifier guidance in GMMs. However, our work focuses on different aspects compared to these studies.
> We will include a detailed comparison in the revised version, as detailed in next bullet.
>
> **Comparison with prior works.** Actually, we have cited and briefly compared with these existing works, which mainly focus on specific classes of distributions like GMMs. In contrast, our main contribution lies in providing a more general theoretical analysis. Nonetheless, we agree that a comparison between our theory--- when applied to specific distributions--- and prior works would further clarify our contributions. We will include a new section in the revised manuscript for comparison:
>
> ''Existing works focus mainly on specific classes of distributions like GMMs, while our work provides a more general theoretical analysis. Below, we compare our findings with prior works when restricted to specific distributions.
>
> In [1], the authors demonstrate that $p_{c | X_0}(c | Y_1^w)\ge p_{c | X_0}(c | Y_1^0)$ holds under specific conditions, while we show that this inequality does not always hold. In addition, [2] argues that guidance can degrade the performance of diffusion models, as it may introduce mean overshoot and variance shrinkage. In contrast, our result shows that guidance can improve sample quality by generating more samples of high quality. Furthermore, [3] shows that classifier guidance can not generate samples from $p(x | c)^{\gamma}p(x)^{1-\gamma}$ for GMMs and establishes its connection to an alternative approach, i.e., the single-step predictor-corrector method, whose effectiveness in this specific setting remains unclear. In contrast, we directly analyze and demonstrate the effectiveness of CFG.''
>
> **Issue of Inception Score (IS).**
> We offer two clarifications:
>
> 1. The reviewer is absolutely correct that IS is not a perfect metric and has known limitations in evaluating sample quality.
>
> 2. We disagree that *''IS is outdated''* and argue that IS is still one of the most widely used and informative metrics for assessing sample quality. The precise evaluation of sample quality remains an open problem. Despite the concerns raised in [4], IS continues to be employed as a key metric in recent works on classifier guidance [5] and classifier-free guidance [6] for diffusion models, due to the absence of an alternative metric that is demonstrably superior to IS. Our choice to consider IS aligns with the original studies that introduced diffusion guidance. In addition, most issues outlined in [4] come from the inaccuracy in the empirical estimation of IS, which doesn't apply for the theoretical consideration, as our results are derived using the true conditional probability.
>
> We will clarify the reasonability for using IS in the revised manuscript:
>
> ''Although some practical limitations of IS have been identified [4], it remains a commonly used metric for evaluating sample quality in the study of diffusion guidance [5,6]. Moreover, in our theoretical analysis, we use the true conditional probability, which addresses the estimation issues discussed in [4].''
>
> **Typos.** We have fixed them in the revised version.
>
> [1] Theoretical Insights for Diffusion Guidance: A Case Study for Gaussian Mixture Models
>
> [2] What does guidance do? A fine-grained analysis in a simple setting
>
> [3] Classifier-Free Guidance is a Predictor-Corrector
>
> [4] A Note on the Inception Score
>
> [5] Diffusion Models Beat GANs on Image Synthesis
>
> [6] Classifier-free diffusion guidance

---

> > ### Comment · Reviewer_qwwj · 2025-04-02
> >
> > Thank you for the response. Regarding IS, most of the papers which you cite are in my opinion outdated. The recent works, particularly those on state-of-the-art class conditional models (or even text-to-image) diffusion models consider Inception Score to be an outdated metric. I strongly advise the authors to consider other metrics.
> >
> > Although I believe that there is a contribution to the literature by your developed, I believe that the paper still falls short of the bar of acceptance, particularly due to its weak empirical evaluations.

---

> > > ### Author Response · Authors · 2025-04-02
> > >
> > > Thanks for your response. We greatly appreciate the time and effort you have dedicated to reviewing our paper.
> > >
> > > **Further response on IS.**
> > > In our previous response, we stated that [5] and [6] introduced classifier(-free) guidance to enhance sample quality by balancing fidelity and diversity, where IS is employed as a key evaluation metric. This motivates our study on the improvement of classifer probability, i.e., $p(c|x)$. In addition, [1], published in ICML last year, also investigated improvements in classifier probability $p(c|z)$ (see Section 3). Moreover, most of the issues with IS discussed in [4] stem from the estimation of $p(c | x)$, which does not applicable for theoretical analysis. Hence, we believe that analyzing the effectiveness of guidance through improvements in classifier probability remains a valuable research direction in this field.
> > >
> > > You said that IS is considered outdated for class-conditional models, including text-to-image models, but did not mention its status in recent works on diffusion guidance. It would be much appreciated if you can provide any recent works that study the effectiveness of diffusion guidance and provide evidence that classifier probability is an inappropriate metric in this context.
> > >
> > > **Further response on experiments.**
> > > We would like to emphasize that this work is theoretical, providing the first theoretical guarantee on the effectiveness of diffusion guidance for general data distributions. While our focus is on theoretical analysis, we have supplemented our findings with empirical validation using both a toy example (GMM) and a real-world dataset (ImageNet). We believe these experiments sufficiently support our theoretical results. For comparison, prior theoretical work on diffusion guidance [1], which was published at ICML last year, conducted experiments only on GMMs. Given this precedent, we believe our empirical evaluations are appropriate for a theoretical study.

---

### Official Review · Reviewer_5U8H · 2025-03-13

**Overall Recommendation:** 4

**Summary:**

This paper gives a novel theoretical analysis of classifier guidance. Whereas prior work focused on special cases, e.g. mixtures of Gaussians and compactly supported distributions, this paper establishes a guarantee under minimal distributional assumptions. Specifically, they consider the functional given by the expected *inverse* classifier probability over generated samples, which is correlated with sample quality, and show that this quantity decreases as the guidance parameter increases. The analysis is short but clever: they utilize th

**Claims And Evidence:**

Yes, please see "Theoretical Claims" for details

**Essential References Not Discussed:**

N/A

**Experimental Designs Or Analyses:**

The experiment was a small-scale evaluation for Gaussian mixtures, and the results appear to be valid and consistent with the theoretical findings.

**Methods And Evaluation Criteria:**

N/A as this is a theory paper

**Other Comments Or Suggestions:**

Please see "Weaknesses" and "Questions for Authors"

**Other Strengths And Weaknesses:**

Strengths:
- This paper's main selling point is that it gives a guarantee for guidance that works for all probability distributions. This is quite exciting, as all prior work focused on specific classes of distributions like Gaussian mixtures, and prior to reading this paper I would not have expected one can prove any interesting, general-purpose result about classifier guidance.
- The key idea is quite clean: (1) the functional they consider is a martingale along the unguided reverse process, so its infinitesimal expected change (which is zero) can be approximated to first order, using Ito's lemma, by an expression depending only on its derivatives in time/space. (2) Likewise, the infinitesimal expected change for the guided reverse process can be approximated to first order using a very similar expression, with some extra terms arising from the guidance term. These extra terms are precisely what give rise to the decrease in expected inverse classifier probability under guidance.

Weaknesses:
- The writing does not do a good job of communicating the main idea behind the calculations in an easy-to-understand manner. I am happy to raise my score if the authors can improve the clarity of the writing
- Unless I'm misunderstanding, the analysis is specific to DDPMs and does not say anything about DDIMs.
- While the authors show the discrete and continuous-time samplers are close using off-the-shelf methods, it's not clear how this closeness can be combined with the main result on expected inverse classifier probability to say something about how that functional behaves under guidance for discrete-time samplers.

**Questions For Authors:**

- Could you comment on how to combine Theorem 3.1 with Theorem 3.6? It's not clear to me that the KL bound should imply anything about the expected inverse classifier probability. I guess if the inverse classifier probability is bounded, then you can use Pinsker's, but such boundedness doesn't seem to be a reasonable assumption.
- Have you tried assessing whether expected inverse classifier probability is actually meaningful in real data? I would imagine it's too large in general to be a useful measure of distributional sample quality

**Relation To Broader Scientific Literature:**

This paper fits into the broader literature on establishing rigorous guarantees for diffusion models.

**Theoretical Claims:**

I checked the correctness of the proof of Theorem 3.1, the main calculation for which appears in Section 4, and believe it to be sound. Please see "Strengths and Weaknesses" for further thoughts on the theory.

---

> ### Author Rebuttal · Authors · 2025-03-31
>
> We thank the reviewer for the constructive feedback! Below, we provide a detailed point-by-point response.
>
> **Explanation of main analysis idea.** In the revised version, we will add the following explanations to better communicate the high-level ideas behind the analysis:
>
> ''**A glimpse of the main analysis idea.** First, this result comes from the key observation that the function of reverse process, $p_{c|X_{t}}(c|X_t)^{\rm -1}$, forms a martingale, as stated in Lemma 3.2, which is established through a careful decomposition of $p_{c|X_{t}}$ and $p_{X_{\tau}|X_{t}}$. Next, the guidance term $s_t(x|c) - s_t(x)$ in classifier-free guidance (CFG) aligns with the direction of $-\nabla p_{c| X_t}(c| x)^{-1} = p_{c| X_t}(c| x)^{-1}[s_t(x|c) - s_t(x)]$, which makes us expect that adding the guidance at time $t$ can decrease $\mathbb{E}\_{x_{\tau} \sim X_{\tau}}\big[p_{c| X_{\tau}}(c| x_{\tau})^{-1} | X_t = x\big]$ for all $\tau \le t$. Finally, to achieve the desired result, particular care must be taken in handling first- and second-order differential terms with respect to $t$ for the process $p_{c| X_{1-t}}(c| Y_t^w)^{-1}$ due to its randomness nature, which is completed in Section 4.2 based on the technique of Ito's formula.''
>
> **Extension to DDIM.** As pointed out by the reviewer, our analysis is specific to DDPMs. In particular, our analysis relies on the martingale property stated in Lemma 3.2, which relies heavily on the property of $p_{X_{\tau}|X_{t}}$ in DDPMs and can not be applied for DDIMs. Extending our framework to DDIMs remains an open question due to the absence of this key property. We will add a remark in Section 3.1 of the revised version to explicitly state this limitation.
>
> **Influence of discretization.** We fully agree that our continuous-time analysis for CFG can not be immediately extended to the discrete-time setting with only small KL divergence error. As the reviewer suggests, if $p(c | x)^{-1}$ is uniformly bounded, one could establish the desired result. However, such an assumption is too strong. Instead, we adopt a weaker condition: we assume that $\mathbb{E}[(p(c | Y_1^w)^{-1}-1)1(p(c | Y_1^w)^{-1} > \tau)]$ is small for some threshold $\tau > 0$. We have also verified this assumption numerically on the Imagenet dataset. We will include the following new result in the revised version:
>
> ''The sampling process (5) with the learning rate schedule (19) satisfies
> \begin{align*}
> \mathbb{E}[p(c | Y_1^w)^{-1}] \le \mathbb{E}[p(c | Y_{\overline\alpha_1}^{w, \mathsf{cont}})^{-1}] + \mathbb{E}[(p(c | Y_{1}^{w})^{-1}-1)1(p(c | Y_{1}^{w})^{-1} > \tau)],
> \end{align*}
> where $\tau$ is defined as the largest value satisfying
> \begin{align*}
> \mathsf{TV}(Y_{\overline\alpha_1}^{w, \mathsf{cont}}, Y_1^w) \le \mathbb{P}(p(c | Y_{1}^{w})^{-1} > \tau).
> \end{align*}
> This further implies the following relative influence from discretization, the ratio between the improvements of $Y_1^w$ and $Y_{\overline\alpha_1}^{w, \mathsf{cont}}$ over $X_{\overline\alpha_1} = Y_{\overline\alpha_1}^{0, \mathsf{cont}}$, obeys
> \begin{align*}
> \frac{\mathbb{E}[p(c | Y_{\overline\alpha_1}^{0, \mathsf{cont}})^{-1}] - \mathbb{E}[p(c | Y_1^w)^{-1}]}{\mathbb{E}[p(c | Y_{\overline\alpha_1}^{0, \mathsf{cont}})^{-1}] - \mathbb{E}[p(c | Y_{\overline\alpha_1}^{w, \mathsf{cont}})^{-1}]}
> \ge 1 - \frac{\mathbb{E}[(p(c | Y_{1}^{w})^{-1}-1)1(p(c | Y_1^w)^{-1} > \tau)]}{\mathbb{E}[p(c | Y_{\overline\alpha_1}^{0, \mathsf{cont}})^{-1}] - \mathbb{E}[p(c | Y_{\overline\alpha_1}^{w, \mathsf{cont}})^{-1}]}.
> \end{align*}
>
> For different values of $\mathsf{TV}(Y_{\overline\alpha_1}^{w, \mathsf{cont}}, Y_1^w)$, we empirically validate the aforementioned assumption on the ImageNet dataset. Here we use $\mathbb{E}[p(c | Y_{1}^{0})^{-1}]-\mathbb{E}[p(c | Y_{1}^{w})^{-1}]$ as an estimate of $\mathbb{E}[p(c | Y_{\overline\alpha_1}^{0, \mathsf{cont}})^{-1}] - \mathbb{E}[p(c | Y_{\overline\alpha_1}^{w, \mathsf{cont}})^{-1}]$. The relative error $\frac{\mathbb{E}[(p(c | Y_{1}^{w})^{-1}-1)1(p(c | Y_1^w)^{-1} > \tau)]}{\mathbb{E}[p(c | Y_{1}^{0})^{-1}]-\mathbb{E}[p(c | Y_{1}^{w})^{-1}]}$ is presented in the following table for various values of the TV distance and $w$. The results indicate that the relative error remains small, particularly for practical choices of $w \ge 1$.
>
> \begin{align*}
> \begin{array}
> \\hline
> \\hline
> \mathsf{TV} & w=0.2 & 0.4 & 0.6 & 0.8 & 1 & 2 & 3 & 4\\\\
> \\hline
> 0.30 & 0.447 & 0.196 & 0.115 & 0.085 & 0.029 & 0.006 & 0.006 & 0.002  \\\\
> 0.10 & 0.440 & 0.194 & 0.114 & 0.085 & 0.029 & 0.006 & 0.005 & 0.002 \\\\
> \\hline
> \end{array}
> \end{align*}
> ''
>
> **Assessment on real dataset.** We have added experiments on the ImageNet dataset to validate our theory. Please refer to our response to Reviewer EKNn of **Experiments on real dataset**.

---

> > ### Comment · Reviewer_5U8H · 2025-04-05
> >
> > Thanks for the new experiments and the thoughtful rebuttal. I have raised my score to a 4.

---

> > > ### Author Response · Authors · 2025-04-05
> > >
> > > Thank you for your acknowledgment and positive evaluation of our work! Your review comments are very helpful in improving the quality of our paper, and we will incorporate your suggestions into the revised manuscript.

---

### Official Review · Reviewer_EKNn · 2025-03-13

**Overall Recommendation:** 3

**Summary:**

In this paper, the authors analyze the effect of diffusion guidance under general data distributions. Their study reveals that guidance does not necessarily improve sample quality in all cases, but it enhances overall sample quality. Specifically, they prove that under the influence of guidance, the proportion of low-quality samples (measured by classifier probabilities) decreases. A toy experiment on the Gaussian Mixture Model provides empirical support for their theoretical analysis.

**Claims And Evidence:**

Each claim in this paper is supported by strong theoretical justification.

**Essential References Not Discussed:**

The citations and comparisons with related works are comprehensive and well-covered, ensuring a thorough contextualization of the proposed approach within the existing literature.

**Experimental Designs Or Analyses:**

The experimental results are reliable—I did not find any obvious issues. However, although this paper primarily focuses on theoretical analysis and proofs, its experimental evaluation is relatively limited. The paper provides only a single case study on a Gaussian Mixture Model, which may not sufficiently demonstrate the practical applicability of the proposed method. Including experiments on more general data distributions would help readers better understand the method and provide stronger inspiration for the research community.

**Methods And Evaluation Criteria:**

This paper provides a sound theoretical analysis of the effect of guidance across general data distributions, offering valuable insights and contributions to the research community. Its findings are likely to inspire further advancements in the field.

**Other Comments Or Suggestions:**

1. Reorganizing the structure of the paper could enhance readability and comprehension. For example, presenting the Toy Experiments as a separate section rather than embedding them within the theoretical analysis would make it easier for readers to follow the logical flow of the paper.

**Other Strengths And Weaknesses:**

**Strengths**
1. The paper is written in a clear and coherent manner.
2. The paper provides a sound and comprehensive analysis of the effect of guidance mechanisms on general data distributions. The analysis is thorough and well-structured, offering valuable insights that could significantly inspire further research in the community.
3. The Toy Example on the Gaussian Mixture Model is a notable highlight of the paper, providing practical support for the theoretical analysis.

**Weaknesses**
1. Although the theoretical analysis in this paper is detailed and comprehensive, the Toy Experiments on the Gaussian Mixture Model alone feel somewhat limited. Conducting experimental analyses on more general data distributions and larger-scale models would provide stronger empirical validation, making the findings more impactful and accessible for the research community and readers.

**Questions For Authors:**

1. How can the theoretical analysis and insights from this paper be used to enhance the capability of the guidance mechanism in practical applications?
2. In my view, $\omega$ is a crucial hyperparameter in classifier-free guidance. When $\omega$ is too small, the generated samples do not align well with the conditions, while an excessively large $\omega$ affects the realism of the generated samples. The experiments conducted on GMM in this paper cover a wide range of $\omega$, and the evaluation metrics improve as $\omega$ increases, which seems inconsistent with observations in practical applications. Could you explain this discrepancy?
3. Could you provide experimental observations on more general benchmarks, such as MNIST or CIFAR-10?

**Relation To Broader Scientific Literature:**

This paper analyzes the effect of guidance on the sampling process in terms of the reciprocal of classifier probabilities, which, to some extent, is conceptually similar to the Inception Score. Moreover, compared to other works, such as Autoguidance [1], which only provide a qualitative analysis of the classifier-free guidance sampling process, this paper offers a detailed and rigorous theoretical analysis, making a more substantial contribution to the understanding of guidance in diffusion models.

[1]: Guiding a Diffusion Model with a Bad Version of Itself

**Theoretical Claims:**

The theoretical analysis in this paper is sound, with detailed and correct proof processes or relevant literature support.

---

> ### Author Rebuttal · Authors · 2025-03-31
>
> Thanks a lot for the reviewer's helpful comments and valuable feedback.
> Below, we provide a point-by-point response, which has also been incorporated into the revised version of our manuscript.
>
> **Experiments on real dataset.**
> Notice that classifier-free guidance (CFG) was originally validated on the ImageNet dataset [1], where Inception Score is typically computed using the Inception v3 classifier [2]. To further support the practical applicability of our method, we have included an additional numerical experiment using a pre-trained diffusion model [3] on the ImageNet dataset, which we believe provides stronger empirical validation than other datasets.
>
> The results demonstrate that guidance improves sample quality by decreasing the averaged reciprocal of the classifier probability rather than achieving uniform improvement across all samples, thereby validating our theory.
> We adopt the guidance scale range in [1] from $0$ to $4$; specifically, in our experiments, we use $w = \{0.2, 0.4, 0.6, 0.8, 1, 2, 3, 4\}$.
> The numerical results are as follows:
>
> \begin{align*}
> \begin{array}{c|cccccccccccc}
> \\hline
> \\hline
> w & 0.2 & 0.4 & 0.6 & 0.8 & 1 & 2 & 3 & 4  \\\\
> \\hline
> {P(p_{c|X_0}(c|Y_1^w) \geq p_{c|X_0}(c|Y_1^0))} & 0.70 & 0.75 & 0.78 & 0.80 & 0.82 & 0.85 & 0.85 & 0.86 \\\\
> -\mathbb{E}[p_{c|X_0}(c|Y_1^w)^{-1}] &-140 & -74 & -47 & -36 & -14 & -3.6 & -3.4 & -2.0 \\\\
> \\hline
> \end{array}
> \end{align*}
>
> **Paper structure.**
> We agree that presenting the numerical experiments as a separate section would enhance readability.
> we will restructure the manuscript by placing the toy experiment and the newly added results on the ImageNet dataset in a separate section following the main results.
>
>
> **Practical implementation.**
> One potential application of our theoretical analysis comes from its implication that guidance may reduce sample quality for a small subset of samples.
> This observation could motivate further research into adaptive guidance mechanisms that ensure a more uniform improvement in sample quality.
> We will include a remark following Theorem 3.1 to discuss the potential implementation of our theory:
>
> ''Theorem 3.1 states that guidance improves the averaged reciprocal of the classifier probability rather than the classifier probability of each individual sample. This suggests that while guidance improves overall sample quality, it may lead to a decline in quality for a small subset of samples. This insight encourages the development of adaptive guidance methods that address this issue and achieve more uniform performance gains, which is a potential practical application of our theory.''
>
>
> **Influence of guidance scale $w$.**
> We agree that extremely large values of $w$ can degrade the performance of CFG.
> However, this phenomenon is consistent with both our theoretical and experimental results.
> In practice, the performance of a diffusion model is typically evaluated based on two key metrics: diversity and sample quality. CFG attains a trade-off between these two metrics, as noted in [1]. In this paper, our main focus is on the influence of guidance on sample quality, particularly in relation to the Inception Score.
> Our results align with practical observations in the sense that the generated samples are more closely adhere to the conditional distribution with larger $w$.
> In addition, previous studies have shown that an extremely large $w$ can severely reduce sample diversity and negatively impact realism. While this is an important consideration for practical applications, it is not the main focus of this work.
> We will add the following remark to avoid confusion:
>
> ''In practice, performance of diffusion models is commonly evaluated by two metrics: diversity and sample quality. This study primarily focuses on the sample quality measured in a similar way as the Inception Score, which increases with $w$. However, prior work [1] has demonstrated that large values of $w$ can significantly reduce sample diversity, leading to unsatisfactory performance in real-world applications.''
>
> **Related works.**
> Reference [4] proposed to use a bad version of the model for guiding diffusion models. We will cite it properly in our revision.
>
>
> [1] Classifier-free Diffusion Guidance
>
> [2] Rethinking the Inception Architecture for Computer Vision
>
> [3] https://github.com/CompVis/latent-diffusion
>
> [4] Guiding a Diffusion Model with a Bad Version of Itself

---

> > ### Comment · Reviewer_EKNn · 2025-04-06
> >
> > Thanks for the author's reply. I'm keeping a positive rating.

---

> > > ### Author Response · Authors · 2025-04-06
> > >
> > > Thank you for your acknowledgment and for your efforts in reviewing our paper. We will revise the manuscript according to your suggestions.

---

### Decision · Program_Chairs · 2025-05-01

**Decision:**

Accept (poster)

**Comment:**

This paper has a split among reviewers. Two of the reviewers like the paper, and view it as a solid initial attempt to better understand the behavior of guidance in diffusion models, which one reviewer describes as otherwise not that easy to study particularly in cases when working with general distributions. The remaining reviewer is worried that the experiments are not realistic or comprehensive enough, and thinks that the theory is not validated experimentally in a tight-enough manner.

In discussion, reviewers did not come to an agreement, and instead decided to emphasize these distinct strengths and weaknesses to back up their judgment, thereby illustrating that these differing views come down to a disagreement about what is important for a paper in this area to have. To this end, I think it is also reasonable to have a paper's contributions emphasize more the theoretical aspects - which precisely pin down intuition coming from practice, even if the setup in which this is done is more restrictive - as opposed to empirical aspects. This is in part because some researchers - for instance those in academia rather than industry - may not have sufficient compute access to perform empirical evaluations to the same degree of comprehensiveness, and may be interested in studying complementary aspects to those with an empirical focus.

In light of this balance, my view is to side with the two positive-score reviewers, and I recommend a weak accept.